# Differential effect of heteronymous feedback from femoral nerve and quadriceps muscle stimulation onto soleus H-reflex

Cristian Cuadra[1,2], Steven L. Wolf[1,3], Mark A. Lyle[1]*

1 Division of Physical Therapy, Department of Rehabilitation Medicine, Emory University School of Medicine, Atlanta, Georgia, United States of America, 2 Exercise and Rehabilitation Sciences Laboratory, School of Physical Therapy, Faculty of Rehabilitation Sciences, Universidad Andres Bello, Santiago, Chile, 3 Senior Research Scientist, Center for Visual and Neurocognitive Rehabilitation, Atlanta VA Health Care System, Atlanta, Georgia, United States of America

* mark.lyle@emory.edu

**Data Availability Statement:** All relevant data are within the paper and its Supporting information files.

**Funding:** Research reported in this publication was supported by the Eunice Kennedy Shriver National

## Abstract

Excitatory feedback from muscle spindles, and inhibitory feedback from Golgi tendon organs and recurrent inhibitory circuits are widely distributed within the spinal cord to modulate activity between human lower limb muscles. Heteronymous feedback is most commonly studied in humans by stimulating peripheral nerves, but the unique effect of non-spindle heteronymous feedback is difficult to determine due to the lower threshold of excitatory spindle axons. A few studies suggest stimulation of the muscle belly preferentially elicits non-spindle heteronymous feedback. However, there remains a lack of consensus on the differential effect of nerve and muscle stimulation onto the H-reflex, and the relation of the heteronymous effects onto H-reflex compared to that onto ongoing EMG has not been determined. In this cross-sectional study, we compared excitatory and inhibitory effects from femoral nerve and quadriceps muscle belly stimulation onto soleus H-reflex size in 15 able-bodied participants and in a subset also compared heteronymous effects onto ongoing soleus EMG at 10% and 20% max. Femoral nerve stimulation elicited greater excitation of the H-reflex compared to quadriceps stimulation. The differential effect was also observed onto ongoing soleus EMG at 20% max but not 10%. Femoral nerve and quadriceps stimulation elicited similar inhibition of the soleus H-reflexes, and these results were better associated with soleus EMG at 20%. The results support surface quadriceps muscles stimulation as a method to preferentially study heteronymous inhibition at least in healthy adults. The primary benefit of using muscle stimulation is expected to be in persons with abnormal, prolonged heteronymous excitation. These data further suggest heteronymous feedback should be evaluated with H-reflex or onto ongoing EMG of at least 20% max to identify group differences or modulation of heteronymous feedback in response to treatment or task.

Institute of Child Health & Human Development of the National Institutes of Health under Award Number K01HD100588 (MAL, https://www.nichd.nih.gov/) and 1R01HD095975-01A1 (SLW, https://www.nichd.nih.gov/), as well as National Institute of Neurological Disorders and Stroke awards 5U01NS086607-05 (SLW, https://www.ninds.nih.gov/funding), U01NS166655 (SLW, https://www.ninds.nih.gov/funding), and 1U01NS102353-01 (SLW, https://www.ninds.nih.gov/funding). The content is solely the responsibility of the authors and does not necessarily represent the official views of the National Institutes of Health. The funders had no role in study design, data collection and analysis, decision to publish, or preparation of the manuscript.

**Competing interests:** The authors have declared that no competing interests exist.

## Introduction

Proprioceptive feedback from human lower limb muscles have widespread excitatory and inhibitory heteronymous spinal projections within the spinal cord that influence motor output of muscles throughout the limb, even those spanning different joints [1–7]. The ability of heteronymous connections to influence motor output of distant muscles suggests a functional role in facilitating interjoint coordination (i.e., neural linkages) [8, 9]. Identifying the relative strength and distribution of excitatory feedback from muscle spindles and inhibitory feedback, which arise primarily from Golgi tendon organs and recurrent motoneuron collaterals, is important for understanding the functional role of heteronymous circuits in normal and impaired movement. However, the unique role of inhibitory circuits is currently difficult to study in isolation between many muscles because peripheral nerve stimulation, the most widely used approach in animals and humans, evokes heteronymous monosynaptic excitation prior to inhibition due to the lower threshold of muscle spindle axons to stimulation when compared to Golgi tendon and motor axons [1, 5, 6, 10, 11]. Thus, alternative methods that can preferentially elicit heteronymous inhibitory feedback from Ib afferents or antidromic activation of motoneuron recurrent collaterals could provide new knowledge about the unique role of inhibitory feedback.

Stimulation of the muscle belly with surface electrodes or intramuscular fine wire elicits heteronymous inhibitory effects (e.g. -Golgi tendon organ and recurrent inhibition) with reduced or absent heteronymous muscle spindle excitation in animals [12, 13] and humans [3, 12, 14]. Of most relevance to the current study, Meunier et al. [14] found that stimulating the vastus lateralis motor point resulted in a much smaller increase in soleus motor unit firing probability compared to femoral nerve stimulation. Stimulation of the quadriceps muscle belly with surface electrodes similarly evoked inhibition of ongoing soleus EMG with similar onset, magnitude, and duration but with much reduced heteronymous excitation when compared to femoral nerve (FN) stimulation [3]. In contrast, Barbeau et al. [15] reported that vastus lateralis motor point stimulation elicited a facilitation of the SOL H-reflex of the same magnitude as femoral nerve stimulation. These findings support further evaluation to examine whether there are differential effects of stimulating the femoral nerve compared to stimulating the quadriceps muscle belly over motor points (referred to as muscle stimulation for the remainder of the text) onto soleus H-reflex or ongoing EMG. This clarification is necessary for examining whether impairments in inhibitory heteronymous feedback are present in patient populations exhibiting exaggerated heteronymous excitation of the soleus H-reflex that may mask the true magnitude of heteronymous inhibition, such as reported in persons with stroke [16, 17].

The purpose of this study was to compare heteronymous excitation and inhibition elicited by femoral nerve and quadriceps muscle stimulation onto the soleus H-reflex in healthy adults. We hypothesize that femoral nerve stimulation, but not quadriceps muscle stimulation, will elicit an increase in soleus H-reflex size when spindle afferent volleys from tibial nerve and femoral nerve stimulation are timed to reach the spinal cord concurrently. We also predict that the magnitude of heteronymous inhibition from femoral nerve and quadriceps muscle stimulation onto soleus H-reflex will be similar. A secondary aim was to compare the heteronymous excitatory and inhibitory effects from femoral nerve and quadriceps muscle stimulation onto ongoing soleus EMG when targeting 10% and 20% of maximal effort in a subset of participants. The motivation to compare the two background SOL EMG intensities is to determine if heteronymous feedback onto ongoing SOL EMG of 10% (i.e., low relative physical effort) can achieve equivalent effects since persons with significant motor impairments may have difficulty consistently holding the 20% EMG effort. Lastly, we examined the correlation between

heteronymous excitation and inhibition from femoral nerve and quadriceps muscle stimulation onto the SOL H-reflex and ongoing EMG. We hypothesize heteronymous effects examined from femoral nerve and quadriceps stimulation onto soleus H-reflex will be strongly correlated with heteronymous effects observed onto ongoing SOL EMG.

## Materials and methods

Fifteen healthy participants (7 males and 8 females, age between 20 to 31 years) with no recent history of lower limb injury or neuromotor disorder participated in this study during 2021. Participants provided written informed consent in accordance with procedures approved by the Emory University Institutional Review Board. Sample size was justified based on pilot data and literature using femoral nerve stimulation [16] suggesting moderate effect sizes. Thus, fourteen participants were found to achieve >83% power to detect a difference of ~5% between nerve and muscle stimulation intensity conditions.

### Equipment

All data were acquired at 2000 Hz using an MP150 acquisition unit (Biopac Systems Inc, Goleta, CA, USA). Surface electromyography (EMG) was recorded from soleus (SOL) and vastus lateralis (VL) muscles with a pair of Ag-AgCl electrodes (2.2 × 3.5 cm; Vermed, Buffalo, NY, USA). Skin was first gently abraded and cleaned with isopropyl alcohol. SOL electrodes were then positioned in the midline of the posterior aspect of the shank just below the gastrocnemius muscle. VL electrodes were positioned on the distal third of the muscle (Fig 1). EMG signals were amplified, and hardware filtered from 10–1000 Hz (AMT-8, Bortec Biomedical Ltd, Canada). Nerve and muscle stimulation was completed with STM100C stimulators (Biopac System Inc, CA, USA).

### Experimental procedures

Participants sat on a dynamometer chair (Humac Norm, CSMI, Inc, Worcester, MA) with the knee and hip flexed at 40˚ and 60˚, respectively. The dynamometer axis of rotation was aligned with the knee joint and a dynamometer pad was firmly positioned against the distal tibia. An ankle immobilizer boot (United Orthopedics, ROM Walker, Fort Wayne, IN, USA) was positioned on their left lower leg. The immobilizer boot stabilized the ankle in a neutral position. The trunk was stabilized using the dynamometer straps.

**Maximal voluntary isometric contractions.** To record maximal SOL EMG, participants were asked to maximally plantarflex focusing on the ankle to minimize knee muscle activity. Participants completed three maximal effort isometric contraction (MVIC) trials of 5 s duration. Verbal encouragement and SOL EMG feedback were provided to facilitate best effort. The SOL EMG was bandpass filtered 10–500 Hz, rectified and a 200 ms moving average was used to identify the peak SOL value for each trial. The mean of the three trials determined maximal SOL EMG and was used to normalize target EMG values to provide visual feedback during conditioning trials.

**Motor threshold for FN and Q stimulation.** Motor threshold was initially identified while the leg was at 90˚ of knee flexion without being attached to the dynamometer to ensure the mechanical effect of the stimulation resulted in knee extension. The FN was stimulated with cathode in the femoral triangle (1 ms pulse width, circular 2.5 cm or 2.2 x 3.5 cm electrodes) generally just medial to the sartorius and approximately 1 cm distal to the inguinal ligament (Fig 1A). The cathode electrode was systematically moved in this area to identify the location that produced an isolated knee extension twitch and minimal hip flexion with the lowest current. The anode was positioned on the posterolateral buttock (7.5 x 13 cm,

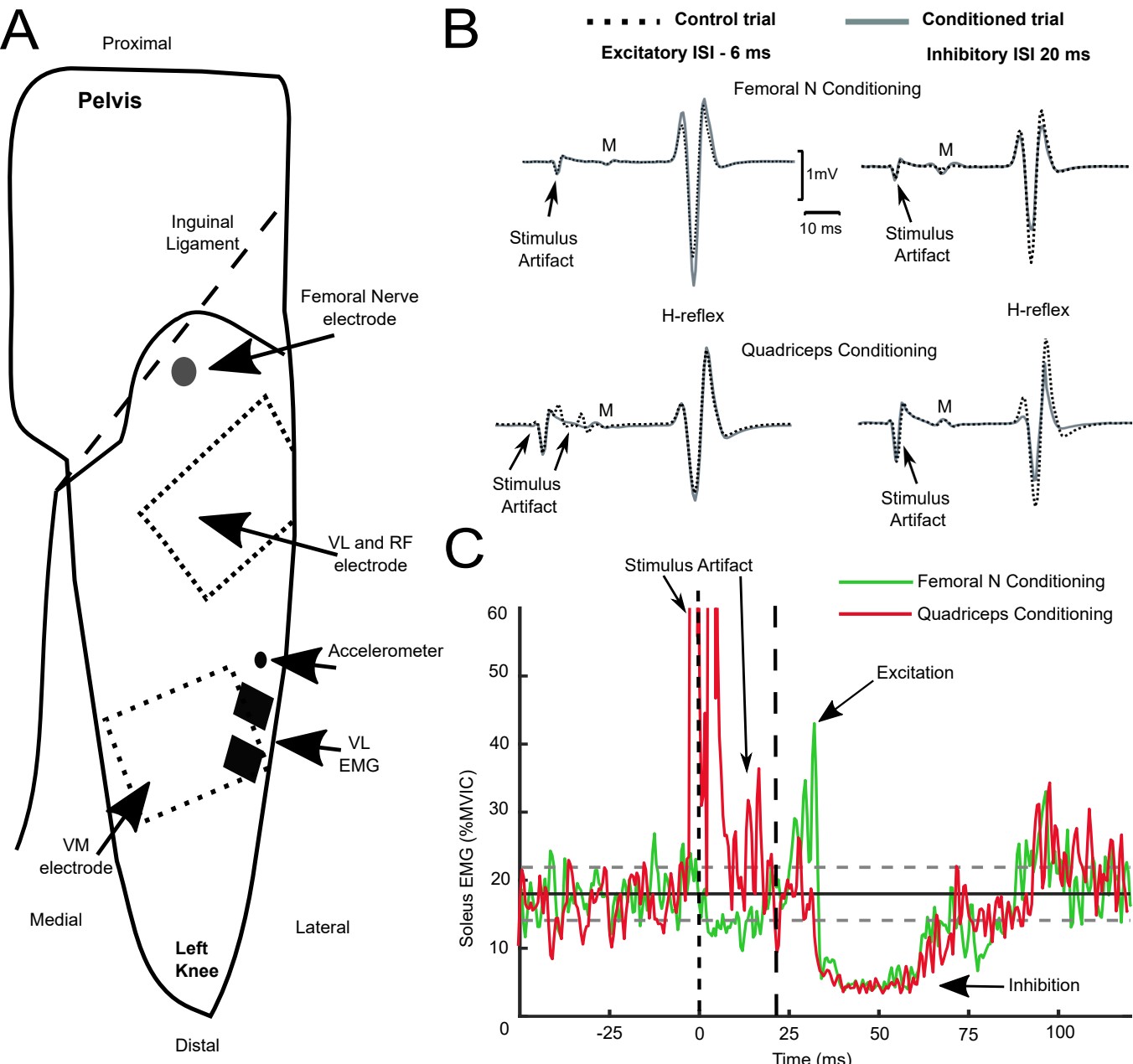

**Fig 1. Schematic of the electrode positioning, and example H-reflex and ongoing EMG waveforms from conditioning trials. A** Shown is the positioning of stimulation and recording electrodes on the left thigh (superior view). The FN cathode electrode is shown on the schematic but the rectangular anode electrode positioned on the posterior buttock is not shown. The dotted rectangles on the thigh correspond to the Q stimulation electrodes. **B** Example waveforms showing control and conditioned H-reflex sizes when FN (top row) was stimulated 6 ms after the tibial nerve (-6, excitatory; left trace) and when FN was stimulated 20 ms prior to tibial nerve (20 ms, inhibitory, right trace). The second row are example control and Q conditioned waveforms (note the Q doublet stimuli were applied 2.5 ms before/after that of FN (i.e. left trace -3.5 and -8.5 ms). All waveforms are the average of 20 trials. **C** Rectified soleus EMG traces (average of 20 trials) from a participant illustrating the heteronymous excitatory and inhibitory effects from femoral nerve (green trace) and quadriceps stimulation (red trace) onto ongoing SOL EMG. The mean background SOL EMG is displayed as the horizontal solid black line, and the dashed horizontal lines represent 1SD above and below the mean. The vertical dashed line at time 0 reflects stimulation timing for FN, whereas the Q stimulation doublet stimuli was at -2.5 and 2.5 ms. The second vertical dashed line represents the period prior to neural effects; note the presence of a Q stimulation artifact just before/after time 0 and two stimulation artifact harmonics that end prior to 18 ms post-stimulus.

ValuTrode, Axelgaard Co., Ltd, CA, USA). One piezoelectric accelerometer (Model A352C65, Model 482C16 Four Channel, PCB Piezotronics, Inc, Depew, NY, USA) was placed on the muscle belly of VL to help confirm motor threshold by identifying the onset of muscle twitches [3]. The criteria for FN stimulation motor threshold was confirmed in the test position (knee flexed at 40 degrees) as the minimal current intensity that produced: 1) a visible M-wave on VL EMG, 2) palpable contraction in VL and VM, 3) and accelerometer onset from VL below ≤20 ms from stimulation [3].

Stimulation of the quadriceps muscles was achieved with two rectangular adhesive electrodes (7.5 x 13 cm ValuTrode, Axelgaard manufacturing Co., Ltd, CA, USA) positioned on motor point areas distally (VM) and proximally (VL-RF) (Fig 1A). Q muscle stimulation motor points were identified with a pen electrode guided by prior anatomical descriptions [18]. Stimulation was a doublet (two unipolar pulses, 50 μs pulse width, 200 Hz) to take advantage of the catchlike property [19, 20] and short pulse width to reduce the likelihood of activating sensory axons [13, 21]. The cathode and anode specification for Q electrodes was based on the configuration that resulted in a concurrent contraction of VM and VL-RF muscles with the lowest intensity. Q motor threshold was confirmed in the test position (knee flexed at 40 degrees) as the minimal current intensity that produced: 1) palpable contraction in VL, RF, and VM.

**Soleus H-reflex and M wave recruitment curve.** To elicit H-reflexes and M-waves, the tibial nerve was stimulated with a single 1 ms pulse in the popliteal fossa with cathode (2.2 x 2.2 cm; Vermed) and proximal anode (4.4 x 4.4 cm) separated by 3–4 cm and oriented along the course of the tibial nerve. Stimulation location was chosen to minimize the SOL H-reflex threshold, maximize the SOL Mmax size, and minimize the TA excitation. H-reflex/M-wave recruitment curves were obtained while participants maintained a plantarflexion contraction between 6% and 10% of their SOL MVIC to ensure stable motoneuron excitability [2, 22, 23]. Visual feedback of SOL EMG and stimulus triggering was provided by EPOCS software (Fig 1). Tibial nerve stimuli were automatically triggered to occur only if the SOL EMG was held in the target EMG range for at least 2–3 s. Tibial nerve stimulation intensity was increased in increments of 1.0–2.5 mA starting from a current below H-reflex threshold and ending when the intensity elicited a maximum M-wave ($M_{MAX}$). Approximately 20 different intensities were used to obtain each recruitment curve, and three EMG responses were averaged at each intensity to determine the H-reflex and M-wave at each intensity. The recruitment curve was used to identify the current needed to produce 50% of $H_{MAX}$, which was the tibial nerve stimulation used when examining heteronymous effects of FN and Q.

**Effect of femoral nerve and quadriceps stimulation onto SOL H-reflex and ongoing SOL EMG.** The general procedure included randomly alternating control and conditioned H-reflex recordings. The control H-reflex (i.e., test H-reflex) was evoked by a tibial nerve stimulation intensity that resulted in the H-reflex size to be around 50% $H_{MAX}$. Conditioned H-reflexes included either Q or FN stimulation paired with tibial nerve stimulation at each of six interstimulus intervals (ISI) described below. Stimulation intensity for Q was 2x motor threshold (46 ± 10.9 mA) and the FN stimulation intensity (1.6 ± 0.2 x motor threshold; range 1.5–2 x motor threshold; 20.7 ± 9.5 mA) was chosen to match the peak torque produced by the 2x motor threshold Q stimulation. Control and conditioned H-reflexes were triggered to occur so long as the SOL activity was within the target range of 6–10% MVIC with a minimum of 8 s between stimulations. The control H-reflex size and M-wave magnitudes were monitored during the experiment; tibial nerve stimulation current was adjusted to keep a consistent M-wave and control H-reflex. A total of 20 trials were recorded for each control and conditioned H-reflex ISI timing. Q and FN conditioning trials were conducted on separate days. Tibial nerve and EMG electrodes were placed in the same location using anatomical and skin related

landmarks as reference [24]. The average days between sessions was 4.2 ± 2.8 days (range: 1–9 days) for 14 of 15 participants. One participant had to travel after the first session so the second session was completed 54 days apart. The Hmax/Mmax ratio for the participants in this study was consistent across days (Day 1: 0.605 ± 0.13 and Day 2: 0.615 ± 0.14, paired t-test: p = 0.43).

The influence of heteronymous proprioceptive feedback from FN and Q stimulation onto SOL H-reflex was examined with 6 ISIs, chosen to include excitatory and inhibitory timings. Stimulation timings are reported in reference to the time of tibial nerve stimulation relative to FN or Q stimulation. Negative ISI intervals reflect tibial nerve stimulation prior to FN or Q stimulation, whereas positive ISI intervals reflect tibial nerve stimulation after FN or Q stimulation.

The negative (-) ISI values of -8, -6, and -4 were chosen to span the typical period of excitatory heteronymous feedback from quadriceps onto SOL H-reflex) [1, 5, 16]. Since the FN and Q muscle belly stimulation electrodes are closer to the spinal cord than the tibial nerve, the tibial nerve is stimulated prior to FN and Q locations when the goal is to have muscle spindle feedback from SOL and quadriceps arrive at SOL motoneurons at the same time (Fig 1B, left traces). The positive (+) ISI values of 0, +20, +60 were chosen to examine inhibitory heteronymous feedback from recurrent inhibition and likely Ib/Golgi tendon organs (Fig 1B, right traces) [1, 6, 16, 25]. Because Q stimulation was a double pulse with 5 ms interpulse interval, the Q stimulation was timed to occur 2.5 ms before and after that of FN for each ISI (i.e., for ISI -8, Q stimulation pulses occurred at -10.5 and -5.5 ms). This timing was selected to account, in part, for differences in the conduction time since FN stimulation is closer to the spinal cord compared to Q motor points; afferent volleys from the proximal VL motor point to the spinal cord has previously been shown to require about 2 ms longer latency compared to FN [14].

After H-reflex conditioning trials were completed, the influence of Q and FN stimulation onto ongoing SOL EMG targeting 10 and 20% MVIC was examined in a subset of 8 participants as described previously [3]. The Q was stimulated at an intensity of 2x motor threshold (i.e., same current used for H-reflex conditioning) when participants maintained SOL EMG at the target 10% for at least 2 seconds. Twenty repetitions were recorded with at least 8 seconds of rest between each repetition. The same procedures were repeated with FN stimulations onto the ongoing 10% SOL MVIC EMG with an intensity that matched the peak torque produced by Q stimulation. The procedures were then repeated while participants targeted 20% SOL MVIC EMG (Fig 1C).

## Data analysis

All data were processed off-line using routines written in MATLAB R2021a (MathWorks, Natick, MA). EMG data were bandpass filtered (0.5–500 Hz, zero phase shift) prior to analysis. To examine whether experimental conditions were consistent within participants, the background SOL EMG activity prior to stimulation and the stimulation evoked peak knee extension torque, and the peak-to-peak SOL M-wave were calculated. Background SOL EMG was computed as the mean of the rectified signal prior to stimulation using a 250 ms window for the H-reflex analyses.

**Effects of femoral nerve and quadriceps stimulation onto SOL H-reflex.** The SOL H-reflex and M-wave sizes were calculated as the peak-to-peak amplitude within individual specified time windows after stimulation (typically, 6–25 ms for the M-wave and 30–50 ms for H-reflex). Excitatory and inhibitory heteronymous effects of FN and Q stimulation onto the SOL H-reflex are reported as a percentage of the control reflex with the following equation:

$$\text{H-reflex size(\% control)} = (\text{conditioned-control})/\text{control} * 100$$

Thus, a negative H-reflex size value indicates reduction of the SOL H-reflex size (i.e., inhibition), whereas a positive value indicates an increase in the H-reflex size (i.e., excitation).

**Effect of femoral nerve and quadriceps stimulation onto ongoing SOL EMG.** Excitatory and inhibitory effects from FN and Q muscle stimulation were evaluated after bandpass filtering (10–500 Hz, zero phase shift) and rectification. The rectified SOL EMG was normalized to each participant's MVIC value and averaged across the 20 repetitions for FN and Q stimulation. The pre-stimulus SOL EMG means and standard deviations (SD) were calculated over a 400 ms period prior to stimulation. To more clearly express intermuscular effects, the mean pre-stimulation SOL EMG was subtracted from the SOL EMG trace, thus delineating the pre-stimulation mean as 0% MVIC and post-stimulation values above and below 0% MVIC excitatory and inhibitory feedback, respectively.

The pre-stimulus background SOL EMG mean and SD for each condition were used to identify excitation and inhibition onset, duration, and magnitudes as described in detail previously [3]. Excitation onset was determined as the time point when the SOL EMG trace exceeded 1SD above the mean for $\geq$ 2 ms. The end of excitation was determined as the time at which the SOL EMG trace returned below the 1 SD line for a period of $\geq$ 2 ms. Only excitatory responses with onset $\geq$ 23 ms were considered as arising from heteronymous Ia facilitation [1, 5]. Inhibition onset and termination were determined as the SOL EMG moving 1SD below the mean background SOL EMG for a period of $\geq$ 2 ms and returning above the 1 SD line for $\geq$ 2 ms. Inhibitory responses were considered in analysis only if the onset was < 45 ms since the fastest transcortical effects could manifest soon thereafter [15]. The durations of excitation and inhibition were calculated as the difference between effect onset and termination. The excitation and inhibition magnitudes were calculated as the area relative to background SOL EMG using trapezoidal numerical integration (trapz in Matlab).

## Statistical analysis

All statistical analyses were performed with R-Studio, version 1.3.1073 (© 2009–2020 RStudio, PBC) and SAS 9.4 (The SAS Institute, Cary, NC). Descriptive statistics are reported in the text and figures as mean ± standard deviation.

To test the primary hypotheses examining heteronymous effects of FN and Q stimulation onto the H-reflex, the MIXED procedure was used to calculate separate two-way repeated-measures ANOVAs for excitatory and inhibitory ISI timings where the variable participant was treated as a random factor, and *stimulation locations* (i.e., FN and Q muscle stimulation) and *ISIs* (i.e., -8, -6, -4 or 0, 20, 60) were considered fixed effects. The *F*-values were computed using a compound symmetry variance-covariance structure and the Kenward-Roger method. Pairwise comparisons were completed with Bonferroni correction. The normality assumption was first inspected with the quantile-quartile plots for each variable and condition separately and evaluated with Shapiro-Wilks test. Friedman's ANOVA was used when ANOVA assumptions of normality and homoscedasticity were not met. In addition, one-sample t-tests (i.e., H-reflex size compared to zero) were used to determine whether FN and Q stimulation elicited significant excitation or inhibition for each ISI.

To examine the heteronymous excitatory and inhibitory effects of FN and Q stimulation onto ongoing SOL EMG, a two-way ANOVA with *stimulation location* (2 levels: FN and Q) × SOL EMG *background level* (2 levels: 10% and 20 SOL %MVIC) was used. Paired post-hoc comparisons were used to examine whether excitatory and inhibitory effects elicited by FN and Q stimulation were significantly different. One-sample t-tests (or Wilcoxon signed rank tests) were used to determine if excitation and inhibition were significantly different from zero. Pearson product moment correlation coefficients were used to examine the

associations between changes in H-reflex size and ongoing EMG due to FN and Q stimulations separately.

To evaluate whether background EMG and M-wave size changed across the experimental conditions, a repeated measures ANOVA with *condition* (2 levels: control and conditioned) x *stimulation location* (2 levels: FN and Q) × ISI (6 levels: -8, -6, -4, 0, 20, 60) was used. To evaluate whether the torque produced by FN and Q stimulation was the same across conditions, a two-way ANOVA with *stimulation location* (2 levels: FN and Q) × *ISI* (6 levels: -8, -6, -4,0, 20, 60) was used. Similarly, consistency in the control H-reflex magnitudes were evaluated with a two-way ANOVA *stimulation location* (2 levels: FN and Q) × ISI (6 levels: -8, -6, -4, 0, 20, 60). Background EMG, M-wave size, control H-reflex magnitude and torque evoked by FN and Q stimulation were all the same across conditions (see S1 File for detailed results). As such, the primary findings of the present study are independent of these factors.

## Results

### Larger heteronymous excitation was elicited with FN compared to Q stimulation

FN stimulation elicited a greater excitation of the SOL H-reflex compared to Q stimulation (Friedman's ANOVA *Chi-squared = 103.92, df = 2, P < 0.001*). The only pairwise difference was found at the –6 ms ISI, where the H-reflex size change due to FN stimulation ($13.95 \pm 9.03\%$ control) was significantly greater compared to Q stimulation ($4.9 \pm 4.53\%$ control, $P = 0.006$, Fig 2A). Furthermore, FN and Q stimulation elicited significant excitation only at the -6 ms ISI (one-sample-test, $P < 0.001$ and $P = 0.002$, respectively). After selecting each

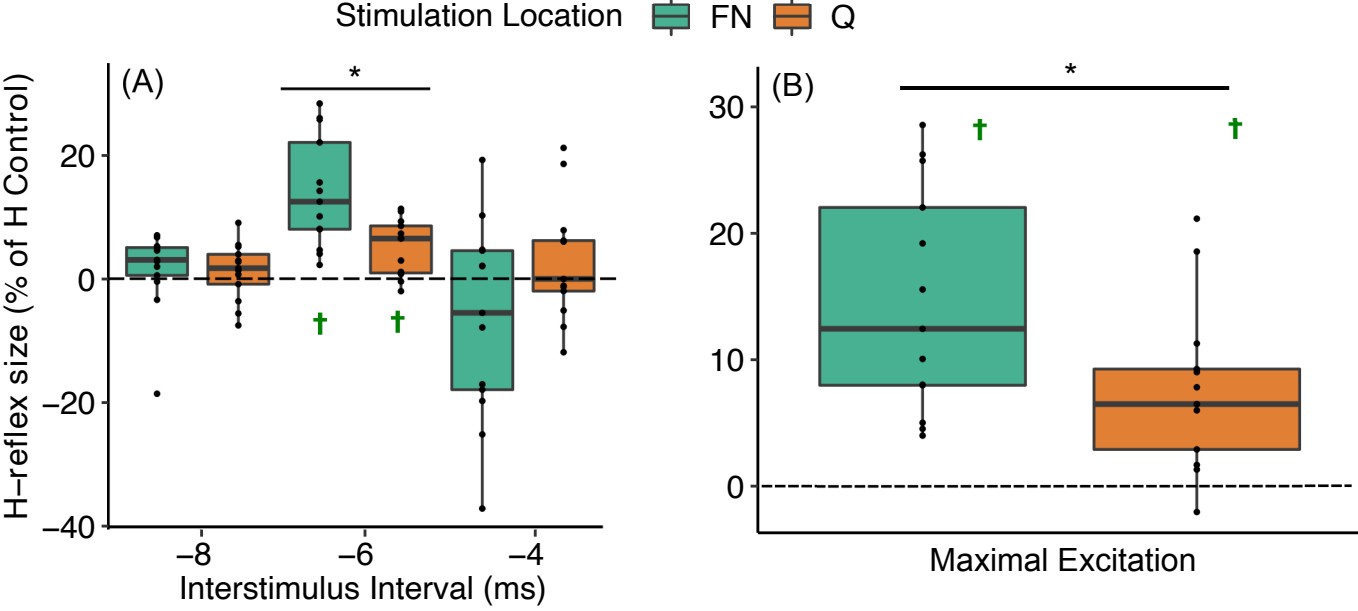

**Fig 2. Heteronymous excitation from femoral nerve (green boxplots) and quadriceps stimulation (orange boxplots) onto the SOL H-reflex. A** Heteronymous excitation for each of the excitatory interstimulus intervals where the largest excitation was observed when the tibial nerve was stimulated 6 ms before the FN or Q (ISI –6, * $P = 0.006$, † significantly different than zero with one-sample t-test, $P < 0.001$). Note that the x-axis ISI values reflect FN stimulation timings. Q stimulation was a doublet with 5 ms interpulse interval such that the first stimuli was 2.5 ms before and after that of FN for each ISI (i.e., for ISI -8, Q stimulation pulses occurred at -10.5 and -5.5 ms). **B** Comparison of participants' maximal heteronymous excitation from all excitatory interstimulus intervals. FN stimulation resulted in larger excitation of the SOL H-reflex compared to Q stimulation. (*$P = 0.011$, †: significantly different than zero with one-sample t-test for all $P < 0.001$). Box and whiskers plots reflect the distribution of the data with the box representing quartiles and the horizontal line the median; each dot represents an individual participant.

participants' largest H reflex size across excitatory ISIs, FN showed larger magnitudes of excitation compared to Q stimulation (14.5 ± 8.87 and 7.68 ± 6.54% control, respectively, Paired t-test: $t_{(12)}$ = 3.005, $P$ = 0.011, Fig 2B).

## FN and Q stimulation resulted in heteronymous inhibition of the SOL H-reflex

FN and Q stimulation elicited significant inhibition of the SOL H-reflex for all inhibitory ISIs (ISIs: 0, 20, 60 ms; One-sample tests: all $P \leq$ 0.008 across stimulation location). There was not a significant interaction between the effects of *stimulation location and ISI* ($F_{2,70}$ = 1.07, $P$ = 0.349, Fig 3). Conversely, there were significant main effects for stimulation location and ISI. FN stimulation elicited greater inhibition compared to Q stimulation (FN: -32.7 ± 18.9 and Q: -23.2 ± -20.9% control H-reflex, $F_{1,70}$ = 8.21, $P$ = 0.0055) and the largest inhibition occurred during the 20 ms ISI timing (0 ms: -24.6 ± 21.2, 20 ms: -36.0 ± 20.6, and 60 ms: -23.3 ± 17.7% control H-reflex, $F_{2,70}$ = 5.91, $P$ = 0.0042). Post hoc pairwise comparisons indicated that FN elicited greater inhibition than Q stimulation at the 0 ms ISI (-32.6 ± 19 and -16.5 ± 20.7% control H-reflex, $P$ = 0.0067), whereas the inhibition was not different between Q and FN at the 20 and 60 ms ISIs (20 ms: -40.0 ± 17.7 and -32.0 ± 22.9% control H-reflex; 60 ms: -25.5 ± 18.6 and -21.0 ± 17.0% control H-reflex, all $P$ >0.05). This finding indicates heteronymous inhibition elicited by Q and FN using 20 and 60 ms ISI timings produce equivalent inhibitory magnitudes.

## Heteronymous effects onto ongoing SOL EMG of 10 and 20% MVIC in subset of 8 participants

The two-way ANOVA examining heteronymous excitation from FN and Q stimulation onto ongoing SOL EMG showed significant main effect for *stimulation location* ($F_{1,21}$ = 13.36, $P$ = 0.0015) and for SOL EMG *background level* ($F_{1,21}$ = 8.31, $P$ = 0.0089) with no interaction. At the 20% SOL EMG target level, FN stimulation resulted in significantly greater excitation compared to Q stimulation ($t_{(21)}$ = 3.96, $P$ = 0.0043, Fig 4A). In addition, FN stimulation

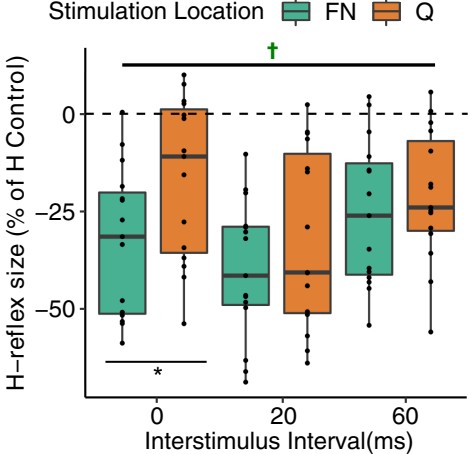

**Fig 3. Heteronymous inhibition from FN and Q stimulation onto SOL H-reflex was observed at each interstimulus interval timing.** (*$P$ = 0.0067, † significantly different than zero with one-sample t-test, all $P \leq$ 0.008). Note that the x-axis ISI values reflect FN stimulation timings. Q stimulation was a doublet with 5 ms interpulse interval such that the first stimuli was 2.5 ms before and after that of FN for each ISI (i.e., for ISI 0, Q stimulation pulses occurred at -2.5 and 2.5 ms). Box and whiskers plots reflect the distribution of the data with the box representing quartiles and the horizontal line the median; each dot represents an individual participant.

resulted in significantly greater excitation onto SOL EMG at the 20% SOL EMG target compared to 10% ($t_{(21)}$ = -3.42, $P$ = 0.016), whereas no differences were found for Q stimulation across SOL EMG levels (Fig 4A). FN and Q stimulation elicited significant excitation for 10% and 20% SOL EMG background levels (One-sample tests: all $P \leq 0.02$).

The two-way ANOVA examining heteronymous inhibition from FN and Q stimulation onto ongoing SOL EMG showed significant main effects for *stimulation location* ($F_{1,21}$ = 7.47, $P$ = 0.0125) and for SOL EMG *background level* ($F_{1,21}$ = 21.51, $P <$ 0.001) with no interaction (Fig 4B). FN stimulation resulted in significantly greater inhibition than Q stimulation (main effect, $P <$ 0.001), but post-hoc pairwise comparisons were not significantly different with Bonferroni correction (P>0.05). FN stimulation resulted in significantly greater inhibition at the 20% SOL EMG background level compared to 10% ($t_{(21)}$ = 4.18, $P$ = 0.0025), whereas inhibition elicited by Q stimulation was greater when targeting 20% compared to 10% SOL EMG

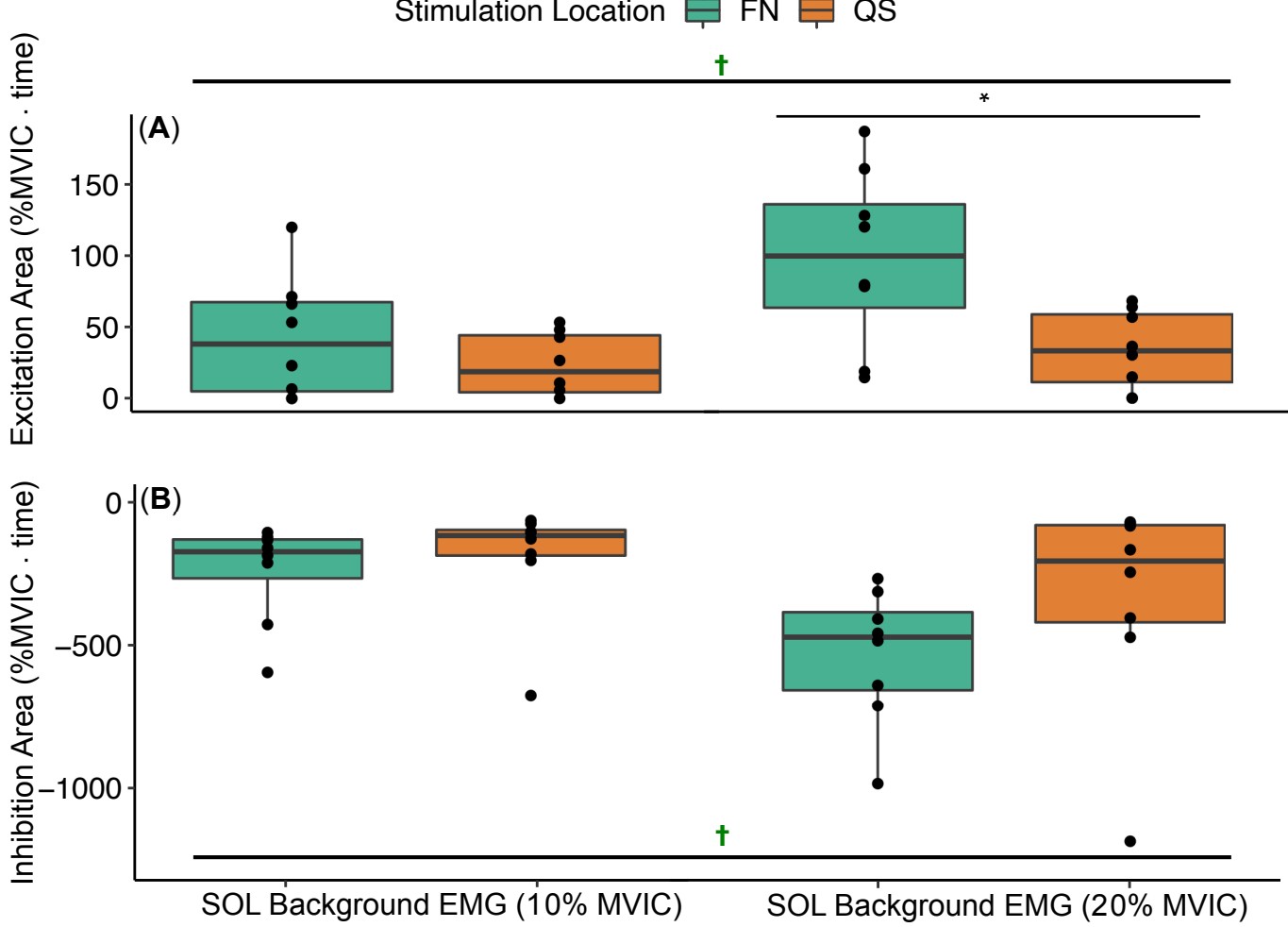

**Fig 4. Heteronymous effects of FN (green) and Q (orange) stimulation onto SOL EMG while targeting 10% SOL EMG (Left pair) and 20% SOL EMG (Right pair). A** Heteronymous excitation was observed from FN and Q onto SOL EMG at 10% and 20% († significantly different than zero with one-sample t-test, all $P \leq$ 0.02). FN stimulation resulted in significantly greater excitation than Q stimulation for the 20% SOL EMG target ($P$ = 0.016). **B** Heteronymous inhibition from FN and Q stimulation onto SOL background EMG at 10% and 20% MVIC. († significantly different than zero with one-sample t-test, all $P \leq$ 0.01). Box and whiskers plots reflect the distribution of the data with the box representing quartiles and the horizontal line the median; each dot represents an individual participant.

on average but the difference was not significantly different. FN and Q stimulation elicited significant inhibition for 10% and 20% SOL EMG background levels (One-sample tests: all $P \leq 0.01$).

## Correlation between heteronymous excitatory effects as measured by H-reflex size and ongoing SOL EMG at 10 and 20% MVIC

The excitation elicited by FN stimulation onto the H-reflex (participants' maximal excitatory H-reflex value across ISIs) was strongly correlated with the excitation elicited during the ongoing SOL EMG targeting 10 and 20% MVIC (r = 0.71, $P$ = 0.047 and r = 0.89, $P$ = 0.002). The excitation elicited by Q stimulation onto the H-reflex (participants' maximal excitatory H-reflex value across ISIs) was weakly correlated with the excitation elicited during SOL EMG targeting 10% (r = 0.32, $P$ = 0.43) and moderately correlated during targeting of 20% SOL MVIC (r = 0.58, $P$ = 0.13). The scatterplots representing the correlation values above can be found in S1 Fig.

## Correlation between heteronymous inhibitory effects as measured by H-reflex size and ongoing SOL EMG at 10 and 20% MVIC

The inhibition elicited by FN stimulation onto the SOL H-reflex (20 ms ISI) was weakly correlated with the inhibition elicited during the ongoing SOL EMG targeting 10% MVIC (r = 0.16, $P$ = 0.7) and moderately correlated with the inhibition elicited during ongoing EMG targeting 20% MVIC (r = 0.52, P = 0.18). The inhibition elicited by Q stimulation onto the H-reflex (20 ms ISI) was strongly correlated with the inhibition elicited during SOL EMG targeting 10% (r = 0.72, $P$ = 0.042) and 20% SOL MVIC (r = 0.82, $P$ = 0.013). The scatterplots representing the correlation values above can be found in S2 Fig.

## Discussion

The purpose of this study was to examine the differential influence of FN and Q stimulation onto the SOL H-reflex using ISI timings anticipated to cause excitation and inhibition. Consistent with the hypotheses and prior work [3, 14], heteronymous excitation elicited by FN stimulation was significantly larger than that elicited by Q stimulation (see Fig 2). Moreover, FN and Q stimulation elicited inhibition of the SOL H-reflex with similar magnitudes for 20 and 60 ISI intervals (see Fig 3). These findings suggest stimulation of the muscle belly can evoke similar heteronymous inhibition with less excitation compared to stimulation of the primary nerve trunk (i.e., FN).

## Heteronymous excitation from FN and Q onto SOL H-reflex and ongoing EMG

Several studies have reported that FN stimulation elicits excitation onto the SOL H-reflex [1, 16, 26, 27] when the tibial nerve is stimulated just prior to the FN such that feedback from muscle spindle axons from both nerves reach the soleus motoneurons near the same time. Based on this prior work, three excitatory ISI timings (i.e., tibial nerve stimulated 4, 6, and 8 ms prior to FN) were compared between FN and Q stimulation. On average, the largest excitatory effect for FN and Q stimulation was observed when the tibial nerve was stimulated 6 ms prior to FN or Q. The magnitude of the heteronymous excitation from FN stimulation onto the SOL H-reflex (around 14%) in the present study is comparable to a prior report [16]. A primary finding of this study was that Q stimulation resulted in about half the heteronymous excitation of that resulting from FN stimulation. These data are consistent with several studies

[3, 12–14] that, taken together, support the utility of muscle stimulation as a way to examine non-spindle related heteronymous circuits in healthy young adults with less or no influence from heteronymous excitation.

While the excitation of the H-reflex from FN stimulation was generally larger than that for Q stimulation within a subject and on average, heteronymous excitation elicited by Q stimulation was evident in some individuals. This finding is consistent with a recent report that also found Q stimulation elicited excitation onto ongoing SOL EMG of 20% MVIC in some participants [3] even though the magnitude was about half the heteronymous excitation elicited by FN stimulation. In contrast to our findings, Barbeau et al. [15] reported FN and localized VL motor point stimulation using 1 ms pulse duration evoked the same heteronymous excitation of the SOL H-reflex in four participants. We speculate that the more localized stimulation of the VL motor point and l ms pulse duration could contribute to the more similar heteronymous SOL H-reflex facilitation compared to the present study which used larger surface electrodes and a doublet of 0.05 ms pulses. While the origin of the excitation cannot be determined using the methods in this study, heteronymous excitation elicited by FN stimulation has previously been attributed to activating muscle spindle axons with monosynaptic projections onto SOL motoneurons [1, 5]. Thus, our working hypothesis is that Q stimulation activates fewer spindle axons and/or activates spindle axons with more temporal dispersion than FN stimulation that results in fewer soleus motoneurons reaching threshold; nonetheless, our findings suggests that quadriceps muscle belly stimulation appears to activate a sufficient population of spindle axons in some individuals. The heteronymous effects from presumed muscle spindle axon activation in some persons observed here are in contrast to the fact that muscle stimulation evokes negligible homonymous H-reflexes [28, 29]. Moreover, muscle stretch (i.e., physiological activation of muscle spindle receptors), but not intramuscular stimulation, evokes heteronymous excitation between cat hindlimb muscles that share strong muscle spindle excitatory feedback [13]. Thus, the heteronymous excitation elicited by Q stimulation in some participants in this study remains a surprising finding though appears consistent with a prior report that also found some individuals exhibit greater central effects of electrical stimulation [30]. The primary factors responsible for those exhibiting excitation due to Q stimulation remain to be identified. We speculate anatomical differences, such as more superficially positioned spindle axons (e.g. see Fig 6 in [30]), could be responsible.

Heteronymous excitatory effects were also examined from FN and Q stimulation onto ongoing SOL EMG at 10% and 20% of MVIC in a subset of 8 participants. In our prior work, we examined heteronymous effects onto SOL EMG with background of 20% MVIC [3]. The goal here was to examine whether the differential effects of FN and Q stimulation could be observed using a lower background EMG magnitude of 10% since this circumstance may be more feasible in some patient populations. Overall, heteronymous excitation was observed from FN and Q onto the 10% SOL EMG. However, the excitatory magnitude elicited by FN stimulation was significantly lower when targeting 10% SOL EMG compared to 20%, and the differential effects of FN and Q stimulation were only observed at the 20% SOL MVIC. Moreover, the correlation was stronger between heteronymous excitation as measured by H-reflex and the 20% SOL EMG target when compared to 10%. Taken together, if a low physical effort is required, these results support evaluating heteronymous excitation from FN and Q stimulation onto the SOL H-reflex rather than onto ongoing 10% SOL EMG. The findings further indicate that the same conclusions are expected when examining heteronymous excitation from FN and Q stimulation onto the SOL H-reflex or when targeting 20% SOL EMG, at least in in young otherwise healthy adults.

## Heteronymous inhibition from FN and Q stimulation onto SOL H-reflex and ongoing EMG

Heteronymous inhibition from FN onto either the SOL H-reflex or ongoing SOL EMG is a common finding that immediately follows the heteronymous excitation, if present, in the time domain [3, 6, 14, 16, 17]. In the present study, inhibition of the SOL H-reflex was observed for FN and Q stimulation at each of the inhibitory ISI timings examined. Overall, FN stimulation resulted in greater inhibition than Q stimulation and the inhibitory magnitudes due to FN stimulation are in close agreement with a prior report [16]. The larger inhibition due to FN stimulation in the present study was primarily due to the 0 ms ISI (tibial nerve and FN or Q stimulation at the same time) for which the Q stimulation elicited inhibition was significantly less. The inhibition elicited by FN and Q stimulation at the 20 and 60 ms ISI timings was not statistically different indicating Q stimulation produces similar inhibitory results as FN stimulation at these time intervals.

The origin of heteronymous inhibition from FN and Q stimulation has been attributed to direct activation of motor axons evoking recurrent inhibition (i.e., activation of Renshaw cells by recurrent collaterals), direct activation of Golgi tendon organ Ib axons and mechanical activation of Golgi tendon organ receptors due to muscle contraction [3, 6, 13, 14]. Golgi tendon organ feedback immediately follows heteronymous excitation in the time domain, and in fact reduces the latter part of heteronymous excitation [1, 7, 11]. While the precise time course transitioning from heteronymous excitation (Ia) to inhibition (Ib) was not systematically examined in this study, the 0 ms ISI timing likely reflects direct activation of Ib axons acting via interneurons that result in inhibitory postsynaptic potentials onto the soleus motoneuron pool and a reduced H-reflex size [1]. The larger inhibition elicited by FN compared to Q stimulation at the 0 ms ISI may reflect a reduced relative activation of Ib axons from the Q muscle stimulation. The heteronymous inhibition observed when FN and Q muscle were stimulated 20 and 60 ms prior to the tibial nerve could arise from several possibilities. In addition to activation of Golgi tendon Ib axons, heteronymous inhibition at the 20 and 60 ms ISI could also arise from recurrent inhibition and mechanical activation of Golgi tendon organs, and the 60 ms ISI could be further influenced by transcortical effects [15]. The prevailing opinion is that the heteronymous inhibition is predominantly due to recurrent inhibition [6, 15, 31]. An alternative possibility is that the stimulation evoked twitch contractions from FN and Q muscle stimulation result in mechanical activation and subsequent firing of Golgi tendon organs [12, 13]. The Q muscles twitch onset for FN and Q stimulation, estimated from an accelerometer, was 10.5 and 7.8 ms after stimulation, respectively. Thus, there appears to be time for Ib afferent firing to have effects at the 20 ms ISI and certainly at the 60 ms ISI, since the twitch evoked force is rising during this period and peaks around 100 ms after stimulation. We chose to examine the 60 ms ISI due to the presumed effects resulting from twitch evoked mechanical activation of Golgi tendon organ receptors, despite prior studies showing heteronymous inhibition to stop around 40 ms [16, 32]. The heteronymous inhibition observed in the present study at the 60 ms ISI could be influenced by the SOL H-reflex recorded during a low-level tonic contraction, whereas prior reports were completed at rest [16, 32]. An alternative explanation for the heteronymous inhibition at the 60 ms ISI could be attributed to presynaptic inhibition [25].

Heteronymous inhibition from FN and Q stimulation was examined onto ongoing SOL EMG held at 10 and 20% of maximal in 8 of the participants. The goal was to determine whether a lower physical effort (i.e., 10% SOL background EMG) could still be used to identify significant inhibition. The FN and Q stimulation elicited significant inhibition when targeting 10% and 20% SOL EMG. Nonetheless, the inhibition magnitude at 20% was larger and most

likely primarily reflects the greater EMG magnitude available to inhibit. This finding is consistent with a prior study that found a significant correlation between the SOL EMG background level and heteronymous inhibition from FN stimulation [33]. As noted for heteronymous excitation and supported by the correlation analysis, the findings suggest that heteronymous inhibition can be observed with 10%; however, targeting 20% background EMG is expected to have a greater discriminant ability across individuals or groups and is more in line with the findings from SOL H-reflex.

## Methodological considerations

The primary results evaluating FN and Q stimulation onto SOL H-reflex was completed while participants produced a low-level active contraction of the SOL. While our results were comparable to prior work, most studies recorded the test and conditioned SOL H-reflexes while the SOL was at rest [1, 16, 32]. We found heteronymous inhibition from FN and Q stimulation onto SOL H-reflex at the 60 ms ISI, whereas studies completed while SOL was at rest found heteronymous to end around 40 ms ISI [16, 32]. Thus, a possible explanation is that the active SOL initial condition could be responsible for the observed difference in heteronymous inhibition duration. An additional methodological consideration was that we examined heteronymous excitation with only a few ISI intervals. We were focused on 6 ms ISI which has been a commonly observed and justified interval [1, 14, 16, 32]. The purpose of this study was to compare FN and Q stimulation and thus the limited ISI timings used were adequate to test the hypothesis. Nonetheless, we acknowledge that the limited excitatory ISI timings make us unable to confirm whether the largest excitatory response was recorded for a given individual. The comparisons and correlations between the heteronymous effects onto the SOL H-reflex and ongoing SOL EMG at two background intensities were completed in a subset of 8 participants. While we feel the findings are helpful for interpreting past and informing future research, a larger sample size would be valuable to further clarify the relations. An additional consideration for future research is the relative influence of the pulse width used for muscle belly stimulation. This study found that stimulating the quadriceps muscle belly using a doublet with 50 μs pulse widths results in less heteronymous excitation when compared to stimulation over the FN with a single 1 ms pulse width stimulus. Several studies have also shown a much-reduced apparent activation of muscle spindle afferents even when applying 1 ms pulse width stimulations to the muscle belly compared to the nerve trunk [14, 28, 29, 34]. While the prior studies support stimulus location as the most likely reason for reduced heteronymous excitation in the present study [14, 28, 29, 34], future work should nonetheless examine the effects of pulse width since heteronymous effects attributed to activating sensory axons have been reported when stimulating other muscle nerves (e.g. medial gastrocnemius, SOL) with 1 ms pulse widths [7, 35].

## Functional implications

Excitatory and inhibitory heteronymous feedback from muscles or motoneurons project extensively onto other lower limb motoneuron pools providing a potential functional role in facilitating motor coordination [4, 5, 14]. While acknowledging our methodical considerations, the heteronymous inhibition in the present and other studies [e.g., 16] highlight that inhibition typically occurs with greater frequency, magnitude, and duration when compared to heteronymous excitation. We propose that the consistent presence of heteronymous inhibition across participants implies, albeit indirectly, that inhibitory feedback between muscles may have a more prominent role in facilitating synergistic activation of muscles in a phase and task-dependent manner [15, 33] compared to heteronymous excitation. Expanding the options

to study heteronymous feedback, particularly the ability to preferentially study specific excitatory and inhibitory spinal circuits, is important for clarifying the functional role of heteronymous feedback. The present study provides evidence that stimulation of the muscle belly may be used to study heteronymous inhibitory feedback more selectively when compared to nerve stimulation at least in healthy neurotypical young adults.

In addition, heteronymous feedback is most commonly examined using either the H-reflex or ongoing EMG. Nevertheless, a comparison of the effects in the same study has not yet been published. Here, we identified that the change in H-reflex size and ongoing SOL EMG at 20% were moderate to strongly correlated, whereas heteronymous effects onto SOL EMG at 10% may reveal significant heteronymous effects but with reduced magnitudes. Importantly, this finding suggests the ability to identify modulation of heteronymous feedback during standing and movement may require ongoing EMG of at least 20% MVIC.

## Conclusions

This study compared the heteronymous effects of FN and Q stimulation onto the SOL H-reflex in healthy young adults. We found that Q stimulation resulted in less heteronymous excitation compared to FN stimulation, whereas similar heteronymous inhibition was elicited with FN and Q stimulation. Comparison of heteronymous effects onto the H-reflex and ongoing SOL EMG of 10 and 20% MVIC revealed that the heteronymous excitation and inhibition, as measured with the change in H-reflex size, were moderately to strongly correlated with 20% SOL EMG. Taken together, if the goal is to preferentially study heteronymous inhibition, then Q stimulation is a preferable choice; the motivation and primary benefit for using muscle stimulation will be to quantify inhibition between lower limb muscles in persons that exhibit large heteronymous excitation that may prevent a valid assessment of heteronymous inhibition [e.g., 16].

## Supporting information

**S1 Fig. Scatter plots showing correlation of heteronymous excitation examined by H-reflex and SOL background EMG 10% on left and 20% on right.**
(EPS)

**S2 Fig. Scatter plots showing correlation of heteronymous inhibition examined by H-reflex and SOL background EMG 10% on left and 20% on right.**
(EPS)

**S1 File. Supporting information.**
(DOCX)

## Acknowledgments

The authors want to thank Dr. Jeremy Hill for his assistance with software configuration, and Creedence Riblett, Christopher Tank, Kelly Jones, and Ignacio Novoa who helped during the data collection sessions.

## Author Contributions

**Conceptualization:** Cristian Cuadra, Steven L. Wolf, Mark A. Lyle.

**Data curation:** Cristian Cuadra, Steven L. Wolf, Mark A. Lyle.

**Formal analysis:** Cristian Cuadra, Steven L. Wolf, Mark A. Lyle.

**Funding acquisition:** Mark A. Lyle.

**Investigation:** Cristian Cuadra, Steven L. Wolf, Mark A. Lyle.

**Methodology:** Cristian Cuadra, Steven L. Wolf, Mark A. Lyle.

**Validation:** Cristian Cuadra, Mark A. Lyle.

**Visualization:** Cristian Cuadra, Steven L. Wolf, Mark A. Lyle.

**Writing – original draft:** Cristian Cuadra, Mark A. Lyle.

**Writing – review & editing:** Cristian Cuadra, Steven L. Wolf, Mark A. Lyle.

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
