## [Decision Letter · Decision Letter 0]

26 May 2023

PONE-D-23-13190Differential effect of heteronymous feedback from femoral nerve and quadriceps muscle stimulation onto Soleus H-reflexPLOS ONE

Dear Dr. Lyle,

Thank you for submitting your manuscript to PLOS ONE. After careful consideration, we feel that it has merit but does not fully meet PLOS ONE’s publication criteria as it currently stands. Therefore, we invite you to submit a revised version of the manuscript that addresses the points raised during the review process.

We look forward to receiving your revised manuscript.

Kind regards,

Tomoyoshi Komiyama, Ph.D

Academic Editor

PLOS ONE

Journal Requirements:

3. We note that Figures 1 in your submission contain copyrighted images. All PLOS content is published under the Creative Commons Attribution License (CC BY 4.0), which means that the manuscript, images, and Supporting Information files will be freely available online, and any third party is permitted to access, download, copy, distribute, and use these materials in any way, even commercially, with proper attribution. For more information, see our copyright guidelines: http://journals.plos.org/plosone/s/licenses-and-copyright.

4. Please upload a copy of Supporting Information Figure/Table/etc. S1 Fig and S2 Fig which you refer to in your text on page 28.

Additional Editor Comments:

Dear Authors,

Your study compared the homogenous effects of FN and Q stimulation on the SOL H reflex in healthy young adults. You found that Q stimulation resulted in less heteronymous excitation compared to FN stimulation, and that FN and Q stimulation induced similar silence suppression. Also, comparison of the heteronymous effect of the H reflex and 10 and 20% MVIC on ongoing SOL EMG showed that heteronymous excitation and inhibition were moderately to strongly correlated with 20% SOL EMG, as measured by changes in H reflex size．

However, I think you should strengthen the reliability of these results by adding as much information as possible.

We thus have some questions and suggestions for the manuscript that you might consider.

I believe these comments will be very helpful in the revision of your study.

Tomoyoshi Komiyama

Reviewers' comments:

Reviewer's Responses to Questions

**Comments to the Author**

1. Is the manuscript technically sound, and do the data support the conclusions?

Reviewer #1: Yes

Reviewer #2: No

2. Has the statistical analysis been performed appropriately and rigorously? 

Reviewer #1: Yes

Reviewer #2: I Don't Know

3. Have the authors made all data underlying the findings in their manuscript fully available?

Reviewer #1: Yes

Reviewer #2: No

4. Is the manuscript presented in an intelligible fashion and written in standard English?

Reviewer #1: Yes

Reviewer #2: Yes

5. Review Comments to the Author

Reviewer #1: OVERALL IMPRESSION

In this study, the authors compare in 15 healthy subjects the excitatory or inhibitory effects observed on the Soleus H-reflex, and the Soleus-EMG in a subgroup of 8 subjects, after electrical stimulation of either the Femoral Nerve or the Quadriceps.

The abstract is well constructed and correctly summarizes the study and its main findings.

In the introduction, the authors raise the question of the technical modalities to study specifically the heteronymous inhibitory circuits of the lower limbs.

The authors refer to previous studies focusing on muscle stimulation rather than on the femoral nerve in order to study more specifically the inhibitory circuits in patients who may present an exaggerated heteronymous excitation of soleus H-reflex.

Hypothesis are well made based on knowledge already acquired.

The figures are clear and accurate. The presentation makes the data easy to understand.

The results are clear and easily readable with the titles of the sub-paragraphs : « larger hetenomymous excitation was elicited with FN compared to Q stimulation (fig 2)», « FN and Q stimulation resulted in heteronymous inhibition of the SOL H-reflex (fig 3)» « heteronymous effects onto ongoing SOL EM of 10 and 20 % MVIC in subset of 8 participants ».

Then the correlations between the data received in H- reflex or EMG at 10 or 20% of the MVIC are given (heteronymous excitatory and inhibitory effects). The data are detailed.

The beginning of the discussion effectively summarizes the key data from this study: Consistent with the hypotheses and prior work [3, 15], heteronymous excitation elicited by FN stimulation was significantly larger than that elicited by Q stimulation (see Fig 2). Moreover, FN and Q stimulation 3elicited inhibition of the SOL H-reflex with similar magnitudes (see Fig 3). These findings suggest stimulation of the muscle belly can evoke similar heteronymous inhibition with less excitation compared to stimulation of the primary nerve trunk (i.e., FN).

The data are compared to previous studies.

Limitations of the study are discussed along with methodological considerations and functional implications.

Materials and methods : The procedures are clearly explained as well as the data analysis and the statistical analysis taking into account the post-hoc analysis.

Conclusions: considering these results the authors propose to use Q stimulation rather than FN stimulation to study heteronymous inhibition in the lower limb. One of the applications would be to use this method in patients with early heteronymous excitation e.g. post-stroke patients (ref 17).

DISCUSSION OF SPECIFIC AREAS FOR IMPROVEMENT

No major issues

Minor issues : I just have minor issues to point out

1. Line 143-144 and fig 1 : when you look at the quadriceps stimulation electrodes on figure 1, you get the impression that they do not exactly match the text: « positionned on motor point areas distally (VM) and proximally –VL-RF). The authors should slightly modify the design of the electrodes position.

2. Line 180 : what do the authors mean by « Q and FN conditioning trials were conducted on separate days » ? How long time betwwen the two days ? Did they compare the Hmax/Mmax ratio between the 2 days ?

3. Line 200 : did the study of the influence of Q and FN stimulation onto EMG occur on the same days ? Could you clarify the time course ?

4. Line 413-415 : « The primary factors responsible for those exhibiting excitation due to Q stimulation remain to be identified. We speculate anatomical differences, such as more superficially positioned spindle axons, could be responsible. »

Can you clarify: Is this just speculation or do you have any studies that can explain this?

Reviewer #2: Comments to the Authors:

The article entitled “Differential effect of heteronymous feedback from femoral nerve and quadriceps muscle stimulation onto Soleus H-reflex” is a research article investigating the effect of femoral nerve (FN) stimulation and quadriceps muscle belly (Q) stimulation onto soleus H-reflex and on-going EMG of soleus. The authors showed that FN stimulation elicited greater facilitation of H-reflex compared to Q stimulation. Similar effect was observed in on-going EMG of 20% maximal voluntary contraction (MVC), but not 10% MVC. Whereas heteronymous inhibition was elicited by FN stimulation and Q stimulation. From these results, the authors claim that Q stimulation is preferable choice to activate inhibitory pathway likely via Ib afferents.

The manuscript was unclear because many of measurement values did not describe in current version. So, it is difficult to support the authors’ claim. I suggest that the authors describe your data more precisely.

General Comments

1. To make it easier for non-experts to understand what was measured and analyzed, I strongly suggest showing representative waveforms of H-reflex and on-going EMG.

2. The authors described in statistical data only instead of raw data.　I felt that many of individual raw data were ambiguous in this study (For example; knee extension torque induced by FN and Q stimulation, control H-reflex amplitude, M-wave amplitude of FN stimulation for MT…). These data will help replicate this study and will be beneficial in clarifying the results. Therefore, I suggest showing these measurement values.

3. The authors confirm the motor threshold (MT) while the knee flexed at 90° position (line 129), however, actual measurement was performed in 40° knee flexion and 60° hip flexion position. It is speculated that this posture difference affects MT. Why did the authors select this position to measure the threshold?

4. The authors described the MT of FN stimulation was the minimal current intensity that produced: 1) a visible M-wave on VL EMG, 2) palpable contraction in VL and VM, 3) accelerometer onset from VL below ≤20 ms from stimulation, and 4) visible motion of the hanging shank after stimulation (line 138) and the MT of Q stimulation was determined as the minimal current intensity that produced: 1) palpable contraction in VL, RF, and VM, with 2) visible motion of the shank after stimulation(line 147). How much the stimulus intensity did the authors actually used? In addition, what was the amplitude of the M-wave during FN and Q stimulation?

5. Different stimulation pulses were used in this study, i.e., FN stimulation was used a single rectangular pulse with 1 ms duration, however, Q stimulation was used a paired pulse with 50 μs duration at 200 Hz. If the authors want to compare difference in stimulation sites, I suggest the authors should align the stimulus waveform. Did you perform that in preliminary experiment? It is unclear from the current results whether it is due to differences in stimulation sites or differences in stimulation waveforms.

6. It was difficult to understand the contents of the figures. What does the boxplot mean? What does the dot? What did the authors compare? Please add a more detailed explanation.

7. The authors compared the results of H-reflex and on-going EMG, however the background EMG level was quite different (H-reflex, 6~10% of MVC [line 156]; on-going EMG, 10 and 20% of MVC [lines 205, 207]). I suggest examining and comparing the effects on the H-reflex when the amount of contraction is changed.

8. The authors claimed that the heteronymous inhibition induced by Q stimulation was similar to that by FN stimulation (Figure 4). However, Q stimulation has a large outlier at each 10% and 20% MVC (Figure 4B). Because the number of participants was small (8 participants, line 201), I feel that the results of one person have a large impact on the author's claim. If exclude this data, it may appear that there is difference in inhibition area elicited by FN stimulation and Q stimulation similar to H-reflex experiment. Therefore, it is difficult to support the authors’ claim from the current result.

6. PLOS authors have the option to publish the peer review history of their article (what does this mean?). If published, this will include your full peer review and any attached files.

Reviewer #1: No

Reviewer #2: No

---

## [Author Response · Author response to Decision Letter 0]

20 Jun 2023

**NOTE: Changes to text noted in each of the responses below are referring to the track changed version of the manuscript.

Reviewer #1: OVERALL IMPRESSION

In this study, the authors compare in 15 healthy subjects the excitatory or inhibitory effects observed on the Soleus H-reflex, and the Soleus-EMG in a subgroup of 8 subjects, after electrical stimulation of either the Femoral Nerve or the Quadriceps.

The abstract is well constructed and correctly summarizes the study and its main findings.

In the introduction, the authors raise the question of the technical modalities to study specifically the heteronymous inhibitory circuits of the lower limbs.

The authors refer to previous studies focusing on muscle stimulation rather than on the femoral nerve in order to study more specifically the inhibitory circuits in patients who may present an exaggerated heteronymous excitation of soleus H-reflex.

Hypothesis are well made based on knowledge already acquired.

The figures are clear and accurate. The presentation makes the data easy to understand.

The results are clear and easily readable with the titles of the sub-paragraphs : « larger hetenomymous excitation was elicited with FN compared to Q stimulation (fig 2)», « FN and Q stimulation resulted in heteronymous inhibition of the SOL H-reflex (fig 3)» « heteronymous effects onto ongoing SOL EM of 10 and 20 % MVIC in subset of 8 participants ».

Then the correlations between the data received in H- reflex or EMG at 10 or 20% of the MVIC are given (heteronymous excitatory and inhibitory effects). The data are detailed.

The beginning of the discussion effectively summarizes the key data from this study: Consistent with the hypotheses and prior work [3, 15], heteronymous excitation elicited by FN stimulation was significantly larger than that elicited by Q stimulation (see Fig 2). Moreover, FN and Q stimulation 3elicited inhibition of the SOL H-reflex with similar magnitudes (see Fig 3). These findings suggest stimulation of the muscle belly can evoke similar heteronymous inhibition with less excitation compared to stimulation of the primary nerve trunk (i.e., FN).

The data are compared to previous studies.

Limitations of the study are discussed along with methodological considerations and functional implications.

Materials and methods : The procedures are clearly explained as well as the data analysis and the statistical analysis taking into account the post-hoc analysis.

Conclusions: considering these results the authors propose to use Q stimulation rather than FN stimulation to study heteronymous inhibition in the lower limb. One of the applications would be to use this method in patients with early heteronymous excitation e.g. post-stroke patients (ref 17).

RESPONSE: We thank Reviewer 1 for his/her encouraging comments/impressions, thorough review, and constructive feedback. We feel that input improved the clarity of this revised manuscript.

DISCUSSION OF SPECIFIC AREAS FOR IMPROVEMENT

No major issues

Minor issues : I just have minor issues to point out

1. Line 143-144 and fig 1 : when you look at the quadriceps stimulation electrodes on figure 1, you get the impression that they do not exactly match the text: « positioned on motor point areas distally (VM) and proximally –VL-RF). The authors should slightly modify the design of the electrodes position.

RESPONSE: Thank you for bringing the misrepresented electrode positioning to our attention. We have significantly modified the figure to show the electrode positioning more clearly with a top-down view. 

2. Line 180 : what do the authors mean by « Q and FN conditioning trials were conducted on separate days » ? How long time between the two days ? Did they compare the Hmax/Mmax ratio between the 2 days ?

RESPONSE: Thank you for the question and opportunity to clarify. In all but 1 participant, the 2 sessions were completed within a week of each other (range 1-9 days; mean 4.2 ± 2.8); One of the participants had sessions separated by 54 days due to unanticipated travel. We have added this information for clarification to the methods (Line 196-198)

We previously reported the H/M ratio average across days and have replaced with session 1 and session 2 values separately which were very close across days, see lines 198-199 (Day 1: 0.605 ± 0.13 and Day 2: 0.615 ± 0.14, paired t-test: p = 0.43)

3. Line 200 : did the study of the influence of Q and FN stimulation onto EMG occur on the same days ? Could you clarify the time course?

RESPONSE: The Q stimulation onto Soleus H-reflex and onto EMG occurred on the same day. The FN stimulation onto the Soleus H-reflex and onto EMG occurred on same day. However, the Q and FN stimulation conditioning sessions were completed on separate days as noted above.

4. Line 413-415 : « The primary factors responsible for those exhibiting excitation due to Q stimulation remain to be identified. We speculate anatomical differences, such as more superficially positioned spindle axons, could be responsible. » Can you clarify: Is this just speculation or do you have any studies that can explain this?

RESPONSE: Thank you for the question. We are unaware of any definitive physiological evidence or specific references. Bergquist et al (2011) have nicely reviewed the spatial distribution of axons in their figure 6. We have added this reference to this speculation sentence. 

Reviewer #2: Comments to the Authors:

The article entitled “Differential effect of heteronymous feedback from femoral nerve and quadriceps muscle stimulation onto Soleus H-reflex” is a research article investigating the effect of femoral nerve (FN) stimulation and quadriceps muscle belly (Q) stimulation onto soleus H-reflex and on-going EMG of soleus. The authors showed that FN stimulation elicited greater facilitation of H-reflex compared to Q stimulation. Similar effect was observed in on-going EMG of 20% maximal voluntary contraction (MVC), but not 10% MVC. Whereas heteronymous inhibition was elicited by FN stimulation and Q stimulation. From these results, the authors claim that Q stimulation is preferable choice to activate inhibitory pathway likely via Ib afferents.

The manuscript was unclear because many of measurement values did not describe in current version. So, it is difficult to support the authors’ claim. I suggest that the authors describe your data more precisely.

RESPONSE: We thank reviewer 2 for his/her time and constructive comments. We feel your important comments and questions have resulted in a better and more clear manuscript. Our goal is to thoroughly describe our methods and results. We hope your concerns are clarified from the comments that follow. 

General Comments

1. To make it easier for non-experts to understand what was measured and analyzed, I strongly suggest showing representative waveforms of H-reflex and on-going EMG.

RESPONSE: We have provided raw data waveforms as illustrative examples as suggested in Figure 1C and 1D.

2. The authors described in statistical data only instead of raw data. I felt that many of individual raw data were ambiguous in this study (For example; knee extension torque induced by FN and Q stimulation, control H-reflex amplitude, M-wave amplitude of FN stimulation for MT…). These data will help replicate this study and will be beneficial in clarifying the results. Therefore, I suggest showing these measurement values.

RESPONSE: We agree with the reviewer’s thoughtful comment. Our goal is to describe our methods and results to allow replication. We provide much of the data relevant to your question is in the supporting information document (i.e. supplementary material). Background activity during the H-reflex testing was consistent across conditions; Torque was not different between Q and FN stimulation conditions (9.2 +/- 2.05 Nm and 8.89 +/- 1.87 Nm, respectively; F1,146 = 3.36, P= 0.07). This information has been added to the supporting information document. Stimulation intensities as multiples of motor threshold were provided (Q=2xMT and FN=1.6) in the original submission. We have also provided stimulation currents in mA (Q stimulation= 46 ± 10.9 mA and FN stimulation: 20.7 ± 9.5 mA) in the revised version of the manuscript (see lines 187-188). We also provide information about the Soleus M-wave and the control H-reflex sizes (FN= 54% Hmax and Q=56% Hmax) in supporting information. 

3. The authors confirm the motor threshold (MT) while the knee flexed at 90° position (line 129), however, actual measurement was performed in 40° knee flexion and 60° hip flexion position. It is speculated that this posture difference affects MT. Why did the authors select this position to measure the threshold?

RESPONSE: We appreciate your observation and acknowledge we wrote this in a confusing manner. We used the 90 degree position initially to assure the stimulation produced knee extension. We verified that the criterion (palpable contraction, visible M-wave, etc.) were met in the testing position. We have updated this paragraph in the methods. 

4. The authors described the MT of FN stimulation was the minimal current intensity that produced: 1) a visible M-wave on VL EMG, 2) palpable contraction in VL and VM, 3) accelerometer onset from VL below ≤20 ms from stimulation, and 4) visible motion of the hanging shank after stimulation (line 138) and the MT of Q stimulation was determined as the minimal current intensity that produced: 1) palpable contraction in VL, RF, and VM, with 2) visible motion of the shank after stimulation(line 147). How much the stimulus intensity did the authors actually used? In addition, what was the amplitude of the M-wave during FN and Q stimulation?

RESPONSE: We thank the reviewer for the question and have provided the stimulation currents in mA in this revised version (see lines 187-188, Quadriceps stimulation: 46 ± 10.9 mA and femoral nerve stimulation: 20.7 ± 9.5 mA). Regarding the M-wave amplitude, unfortunately, we do not have this data. We did not record VL EMG during the quadriceps stimulation in part because of the large stimulus artifact from the doublet, making M-wave calculation difficult to achieve. We were also challenged to properly place VL surface EMG electrodes due to the large rectangular size relative to the morphology of some participants’ thighs. We agree having a comparison of M-wave size would be ideal; as an alternative, we controlled for the mechanical output of the muscle in response to the stimulation as knee extension torque (values provided in supporting information).

5. Different stimulation pulses were used in this study, i.e., FN stimulation was used a single rectangular pulse with 1 ms duration, however, Q stimulation was used a paired pulse with 50 μs duration at 200 Hz. If the authors want to compare difference in stimulation sites, I suggest the authors should align the stimulus waveform. Did you perform that in preliminary experiment? It is unclear from the current results whether it is due to differences in stimulation sites or differences in stimulation waveforms.

RESPONSE: We thank the reviewer for the question. The muscle stimulation parameters were chosen to reduce likelihood of activation of sensory axons as referenced in methods. The femoral nerve stimulation parameters were chosen based on the traditional approach used in many prior studies (e.g., Meunier et al. 1990; Dyer et al. 2014); thus, we feel this comparison is a valid test to address the study purpose. Future work should examine whether the effects observed due to muscle stimulation would differ using other stim parameters.

 While we have not evaluated quadriceps muscle stimulation with 1 ms pulse width, sensory contributions from muscle stimulation evoked contractions has previously been reported to be much smaller compared to nerve stimulation in the quadriceps (Bergquist et al. 2012) and H-reflex responses are not readily elicited with muscle stimulation (e.g. Bergquist et al 2011; Bergquist et al. 2012; Nakagawa et al., 2020). Nonetheless, we agree future work should examine the relative influence of pulse width in the methodological considerations section; In addition, we include and acknowledge evidence in which stimulation with 1 ms pulse widths of other muscle nerves (medial gatsroc, soleus) can elicit heteronymous responses attributed to sensory axons (e.g,. Horslen et al. 2017; Pierrot-Deseilligny et al. 1981). Please see end of methodological considerations paragraph in discussion (Lines 521-532)

6. It was difficult to understand the contents of the figures. What does the boxplot mean? What does the dot? What did the authors compare? Please add a more detailed explanation.

RESPONSE: We thank the reviewer for identifying this oversight. We have included a description in each of the figure captions describing what the boxplot and dots represent (i.e., box represents quartiles of data, horizontal line reflects median, and each dot is individual participant data).

7. The authors compared the results of H-reflex and on-going EMG, however the background EMG level was quite different (H-reflex, 6~10% of MVC [line 156]; on-going EMG, 10 and 20% of MVC [lines 205, 207]). I suggest examining and comparing the effects on the H-reflex when the amount of contraction is changed.

RESPONSE: We appreciate the comment. The H-reflex size is a common dependent variable used to examine the strength of a reflex pathway between two muscles. The H-reflex size recorded during a low level of background EMG (6-10%), as used in the present study, is advocated and a common approach used by many researchers (Knikou 2008; Theodosiadou et al. 2023; Stein and Thompson, 2006). Ongoing EMG is also a common dependent variable used when examining the strength of a reflex pathway between muscles. The influence of heteronymous feedback from femoral nerve onto ongoing soleus emg has been evaluated at 20% MVC (Lyle et al, 2022) and 30% MVC (Dyer et al 2014; Meunier et al. 1996). The motivation to use 10% MVC in this study was to identify whether a lower physical effort (10 vs 20% MVC) could be used to quantify excitatory and inhibitory feedback. A lower physical effort of 10% would be preferable for patient populations with residual muscle weakness. These patients can have a hard time sustaining a 20% or greater MVC. 

We feel our comparison of the heteronymous effects from femoral nerve/quadriceps muscle stimulation onto the H-reflex size and ongoing EMG as described in this study are consistent with best practices. While H-reflex size could be evaluated at higher background levels in future studies, we feel that our methodological approach should adhere to best practice principles. Apologies if we do not fully understand your comment.

8. The authors claimed that the heteronymous inhibition induced by Q stimulation was similar to that by FN stimulation (Figure 4). However, Q stimulation has a large outlier at each 10% and 20% MVC (Figure 4B). Because the number of participants was small (8 participants, line 201), I feel that the results of one person have a large impact on the author's claim. If exclude this data, it may appear that there is difference in inhibition area elicited by FN stimulation and Q stimulation similar to H-reflex experiment. Therefore, it is difficult to support the authors’ claim from the current result.

RESPONSE: We appreciate the comment and agree heterogeneity with a small sample can influence results. In relation to H-reflex comparisons, the inhibition was similar between 20 and 60 ms ISI (though inhibition from FN stimulation was larger than Q at 0 ms ISI). Therefore, we have made this explicit in first paragraph of discussion. 

In relation to heteronymous effects onto EMG, we have added a sentence in the results expanding upon significant main effect of stimulation location. Femoral nerve inhibition was greater than quadriceps per significant main effect. The pairwise comparisons were not different statistically possibly due to sample size and because 1 participant had larger inhibition relative to the sample for both FN and Q. We acknowledge in the discussion (methodological considerations paragraph) that the small sample size is a limitation. We do state in the discussion that overall FN caused larger inhibition than Q (see line 461-467); the statistical analysis revealed the difference was due to the 0 ms ISI with no differences found between 20 and 60 ms ISI.

---

## [Decision Letter · Decision Letter 1]

18 Jul 2023

PONE-D-23-13190R1Differential effect of heteronymous feedback from femoral nerve and quadriceps muscle stimulation onto Soleus H-reflexPLOS ONE

Dear Dr. Lyle,

Thank you for submitting your manuscript to PLOS ONE. After careful consideration, we feel that it has merit but does not fully meet PLOS ONE’s publication criteria as it currently stands. Therefore, we invite you to submit a revised version of the manuscript that addresses the points raised during the review process.

We look forward to receiving your revised manuscript.

Kind regards,

Tomoyoshi Komiyama, Ph.D

Academic Editor

PLOS ONE

Dear authors,

Thank you for your submitting your revised manuscript.

I think it is easier to understand than the previous version.

However, Reviewer 2 had additional comments.

Please answer these questions as listed below.

Tomoyoshi Komiyama

Reviewers' comments:

Reviewer's Responses to Questions

**Comments to the Author**

1. If the authors have adequately addressed your comments raised in a previous round of review and you feel that this manuscript is now acceptable for publication, you may indicate that here to bypass the “Comments to the Author” section, enter your conflict of interest statement in the “Confidential to Editor” section, and submit your "Accept" recommendation.

Reviewer #1: All comments have been addressed

Reviewer #2: All comments have been addressed

2. Is the manuscript technically sound, and do the data support the conclusions?

Reviewer #1: Yes

Reviewer #2: Yes

3. Has the statistical analysis been performed appropriately and rigorously? 

Reviewer #1: Yes

Reviewer #2: Yes

4. Have the authors made all data underlying the findings in their manuscript fully available?

Reviewer #1: Yes

Reviewer #2: Yes

5. Is the manuscript presented in an intelligible fashion and written in standard English?

Reviewer #1: Yes

Reviewer #2: Yes

6. Review Comments to the Author

Reviewer #1: I thank the authors for answering my questions and requests for clarification.

The manuscript is now quite clear and deserves publication in an international journal.

Reviewer #2: Comments to the Authors:

The authors have made many of the changes I suggested and the quality of the manuscript is improved, but several minor issues remain:

General Comments

The authors mentioned that excitation and inhibition elicited by FN and Q stimulation was detected by using pre-stimulus background EMG and ±1SD values (line 245~). In addition, the onset latency of excitation/inhibition was set based on the previous reports. However, I think that there are many cases where excitation/inhibition are induced other than the specified latency such as Figure 1C (Quadriceps conditioning; inhibitory like response at -30 ms, facilitatory like response at 15~20 ms). What did the authors do if the reflex was not induced at the desired latency? Did the authors measure several times? Or increase the number of stimulation and averaged? Was 1SD the appropriate for this study (Lines 247, 252)?

Specific Comments

Line 315, 335, 357: What is “a participant”? Should explain more precisely.

Line 482: Does “GTO” mean “Golgi tendon organs”?

Line 422: I cannot follow the author’s claim that Ia afferents contribute the facilitation in the current version. Please describe more why the authors considered Ia afferents contribute to the facilitation in terms that non-experts can understand.

Line 462: How did the previous reports reveal the contribution of Ib afferents to the inhibition? Please describe more why the authors considered the recurrent inhibition and Ib inhibition contribute to the inhibition in terms that non-experts can understand.

Figure 1A: The configuration of stimulation electrodes is difficult to understand from this picture. Does this picture only show the cathode? Where is the anode?

Figure 1B: The authors showed the representative waveforms elicited by FN stimulation. Please show additional waveforms elicited by Q stimulation.

Figure 1B: I think there is a stimulus artifact due to the conditioning stimulation (especially, left waveform; ISI -6 ms). Is it correct?

Figure 2A: I think the horizontal axis is not correct. The authors should describe that the ISI is different for the FN stimulation and Q stimulation as in lines 207~212.

Figure 4: Please add the representative waveforms for 10% and 20% MVIC conditions.

7. PLOS authors have the option to publish the peer review history of their article (what does this mean?). If published, this will include your full peer review and any attached files.

Reviewer #1: No

Reviewer #2: No

---

## [Author Response · Author response to Decision Letter 1]

25 Jul 2023

Review Comments to the Author

Reviewer #1: I thank the authors for answering my questions and requests for clarification.

The manuscript is now quite clear and deserves publication in an international journal.

 RESPONSE: We thank reviewer 1 for their constructive review.

Reviewer #2: Comments to the Authors:

The authors have made many of the changes I suggested and the quality of the manuscript is improved, but several minor issues remain:

 RESPONSE: We appreciate the positive feedback and your constructive review.

General Comments

The authors mentioned that excitation and inhibition elicited by FN and Q stimulation was detected by using pre-stimulus background EMG and ±1SD values (line 245~). In addition, the onset latency of excitation/inhibition was set based on the previous reports. However, I think that there are many cases where excitation/inhibition are induced other than the specified latency such as Figure 1C (Quadriceps conditioning; inhibitory like response at -30 ms, facilitatory like response at 15~20 ms). 

 RESPONSE: We thank the reviewer for the question. In our methods, we indicate the latency criterion (excitation onset >23 ms and inhibition onset <45 ms post the electrical stimulation) we used that is based on conduction times previously established by other authors. An additional criterion was that the excitatory and inhibitory responses had to last at least 2 ms. The “inhibitory like response at -30 ms” did not last 2 ms. The “quadriceps conditioning facilitatory like response at 15-20 ms” the reviewer questioned in Fig 2C is a stimulus artifact harmonic that occasionally happens. The three peaks you note during this region do not stay above the 1SD line for 2 ms and occur prior to the earliest possible neural effects (3rd peak at 17 ms) from activating muscle spindle axons. We have added new labels on the figure and text in the caption describing the stimulation artifact. 

What did the authors do if the reflex was not induced at the desired latency? Did the authors measure several times? Or increase the number of stimulation and averaged? Was 1SD the appropriate for this study (Lines 247, 252)?

 RESPONSE: Facilitatory and inhibitory reflex responses were quantified as described in the methods (i.e., excitatory: soleus EMG exceeding 1 SD above the mean background for at least 2 ms so long as after 23 ms from stimulation). We are not sure exactly what is meant by “measured several times” or “increase the number of stimulation and averaged”? The analyses were completed on the rectified soleus EMG as an ensemble average of 20 repetitions. We feel the 1 SD and 2 ms period was appropriate. 

 For reference from methods (lines 254-261): “Excitation onset was determined as the time point when the SOL EMG trace exceeded 1SD above the mean for ≥ 2 ms. The end of excitation was determined as the time at which the SOL EMG trace returned below the 1 SD line for a period of ≥ 2 ms. Only excitatory responses with onset ≥ 23 ms were considered as arising from heteronymous Ia facilitation [1, 5]. Inhibition onset and termination were determined as the SOL EMG moving 1SD below the mean background SOL EMG for a period of ≥ 2 ms and returning above the 1 SD line for ≥ 2 ms. Inhibitory responses were considered in analysis only if the onset was < 45 ms since the fastest transcortical effects could manifest soon thereafter [15].”

Specific Comments

Line 315, 335, 357: What is “a participant”? Should explain more precisely.

 RESPONSE: The comment appears to refer to our figure captions where we use the phrase “each dot represents a participant.” We use the term participant to indicate a person that was recruited and participated in this study. We have modified the phrase to read “each dot represents an individual participant” to hopefully make clear that the data point dots represent separate persons that participated in the study.

Line 482: Does “GTO” mean “Golgi tendon organs”?

 RESPONSE: Thank you for identifying the use of the abbreviation without defining this clearly in the text. We have written Golgi tendon organ for all occurrences instead of the abbreviation on lines 62 and 503.

Line 422: I cannot follow the author’s claim that Ia afferents contribute the facilitation in the current version. Please describe more why the authors considered Ia afferents contribute to the facilitation in terms that non-experts can understand.

 RESPONSE: The comment is in a paragraph of the discussion that is discussing potential reasons for some participants exhibiting excitation due to quadriceps stimulation. The most widely accepted origin of excitation is from muscle spindle feedback (see Line 437: “Heteronymous excitation elicited by FN stimulation arises from activating muscle spindle axons with monosynaptic projections onto SOL motoneurons [1, 5].” We have added some additional text in the paragraph referenced in your comment and in several other paragraphs (e.g., lines 208-210; 415-417) to help explain for non-experts. 

Line 462: How did the previous reports reveal the contribution of Ib afferents to the inhibition? Please describe more why the authors considered the recurrent inhibition and Ib inhibition contribute to the inhibition in terms that non-experts can understand.

 RESPONSE: The papers cited are a mix of animal and human studies; we feel the methodological details are difficult to summarize for non-experts due to several approaches (e.g., nerve stimulation and recording from motoneurons in decerebrate cat; controlled pairwise muscle stretches of 2 muscles in the decerebrate cat) and beyond the scope of this paper. We feel the main ideas that apply here are that Golgi tendon organ Ib sensory axons or motor axons (evoke recurrent inhibition) can be activated directly by the stimulation current and the twitch contraction can mechanically activate Golgi tendon organ receptors. We have added text we hope helps non-experts better understand the distinction between effects elicited by current acting on sensory/motor axons and mechanical activation of Golgi tendon organ receptors (e.g. Lines 480-489).

Figure 1A: The configuration of stimulation electrodes is difficult to understand from this picture. Does this picture only show the cathode? Where is the anode?

 RESPONSE: Sorry for the confusion. We specifically added a top-down view to make the electrode configuration more clear. The cathode electrode is shown for the femoral nerve electrode but the anode is positioned on the posterior buttock and not shown. We have added a note in the caption about not including the anode here. Two electrodes show the positioning of quadriceps muscle belly stimulation electrodes on the figure and are described in the caption as dotted lines. The cathode and anode specification for the quadriceps was based on the lowest intensity that resulted in a contraction of both VM and VL/RF. We have added this text to the methods section. 

Figure 1B: The authors showed the representative waveforms elicited by FN stimulation. Please show additional waveforms elicited by Q stimulation.

 RESPONSE: We have added Q stimulation waveforms to Figure 1B.

Figure 1B: I think there is a stimulus artifact due to the conditioning stimulation (especially, left waveform; ISI -6 ms). Is it correct?

 RESPONSE: The figure shows a stimulus artifact from the tibial nerve stimulation. There is not a clear stimulus artifact seen on this waveform from the FN stimulation which is typical. Please see the Quadriceps stimulation excitatory waveform has a visible stimulation artifact since the stimuli occur just after the tibial nerve stimulation.

Figure 2A: I think the horizontal axis is not correct. The authors should describe that the ISI is different for the FN stimulation and Q stimulation as in lines 207~212.

 RESPONSE: We have added text to the caption to make clear that Q stimuli were applied 2.5 ms before and after the FN single stimulus for each ISI.

Figure 4: Please add the representative waveforms for 10% and 20% MVIC conditions.

 RESPONSE: Since representative waveforms illustrating the FN and Q stimulation conditions at 20% MVIC are shown in Fig 1C, we prefer to not include more waveforms here.

---

## [Decision Letter · Decision Letter 2]

1 Aug 2023

Differential effect of heteronymous feedback from femoral nerve and quadriceps muscle stimulation onto Soleus H-reflex

PONE-D-23-13190R2

Dear Dr. Lyle,

We’re pleased to inform you that your manuscript has been judged scientifically suitable for publication and will be formally accepted for publication once it meets all outstanding technical requirements.

Kind regards,

Tomoyoshi Komiyama, Ph.D

Academic Editor

PLOS ONE

Additional Editor Comments (optional):

Dear authors,

Thank you for submitting your revised manuscript.

I think it was much easier to understand than the original manuscript.

I am satisfied with the responses and the edits, I am happy to accept this manuscript.

The authors have replied to my remaining questions satisfactorily from two reviewers.

Therefore, I have no further comments to make, all of my previous concerns were adequately addressed.

This manuscript will be satiating the reader's interest.

Tomoyoshi Komiyam

Reviewers' comments:

Reviewer's Responses to Questions

**Comments to the Author**

1. If the authors have adequately addressed your comments raised in a previous round of review and you feel that this manuscript is now acceptable for publication, you may indicate that here to bypass the “Comments to the Author” section, enter your conflict of interest statement in the “Confidential to Editor” section, and submit your "Accept" recommendation.

Reviewer #2: All comments have been addressed

2. Is the manuscript technically sound, and do the data support the conclusions?

Reviewer #2: Yes

3. Has the statistical analysis been performed appropriately and rigorously? 

Reviewer #2: Yes

4. Have the authors made all data underlying the findings in their manuscript fully available?

Reviewer #2: Yes

5. Is the manuscript presented in an intelligible fashion and written in standard English?

Reviewer #2: Yes

6. Review Comments to the Author

Reviewer #2: Comments to the Authors:

I thank the authors for addressing my previous concerns. Below are my last comments:

Specific Comments

Line 325, 348, 370: “Each dot represents an individual participant”

I think the sentence “mean value of excitation and/or inhibition for each participant” is more appropriate rather than “an individual participant”.

Figure 1: The term “Stimulus artifact” is unclear. Please describe what stimulation caused the artifact (i.e., tibial nerve stimulation, femoral nerve stimulation, quadriceps stimulation). I think that showing the timing of stimulation will help the reader understand the experimental protocol.

7. PLOS authors have the option to publish the peer review history of their article (what does this mean?). If published, this will include your full peer review and any attached files.

Reviewer #2: No

---

## [Editor Report · Acceptance letter]

4 Aug 2023

PONE-D-23-13190R2 

Differential effect of heteronymous feedback from femoral nerve and quadriceps muscle stimulation onto Soleus H-reflex 

Dear Dr. Lyle:

I'm pleased to inform you that your manuscript has been deemed suitable for publication in PLOS ONE. Congratulations! Your manuscript is now with our production department. 

Kind regards, 

on behalf of

Dr. Tomoyoshi Komiyama 

Academic Editor

PLOS ONE